# Characterization of the Native Disulfide Isomers of the Novel χ-Conotoxin PnID: Implications for Further Increasing Conotoxin Diversity

**DOI:** 10.3390/md21020061

**Published:** 2023-01-19

**Authors:** Michael J. Espiritu, Jonathan K. Taylor, Christopher K. Sugai, Parashar Thapa, Nikolaus M. Loening, Emma Gusman, Zenaida G. Baoanan, Michael H. Baumann, Jon-Paul Bingham

**Affiliations:** 1Department of Molecular Biosciences and Bioengineering, College of Tropical Agriculture and Human Resources, University of Hawai’i, Honolulu, HI 96822, USA; 2School of Pharmacy, Pacific University, 222 SE 8th Ave, Ste. 451, Hillsboro, OR 97123, USA; 3Department of Chemistry, Lewis & Clark College, 615 S Palatine Hill Road, Portland, OR 97219, USA; 4Department of Biology, College of Science, University of the Philippines Baguio, Baguio City 2600, Philippines; 5Designer Drug Research Unit, Intramural Research Program, National Institute on Drug Abuse (NIDA), National Institutes of Health (NIH), 333 Cassell Drive Suite 4400, Baltimore, MD 21224, USA

**Keywords:** peptide, toxins, conotoxins, isomers, structure, monoamine transporters

## Abstract

χ-Conotoxins are known for their ability to selectively inhibit norepinephrine transporters, an ability that makes them potential leads for treating various neurological disorders, including neuropathic pain. PnID, a peptide isolated from the venom of *Conus pennaceus*, shares high sequence homology with previously characterized χ-conotoxins. Whereas previously reported χ-conotoxins seem to only have a single native disulfide bonding pattern, PnID has three native isomers due to the formation of different disulfide bond patterns during its maturation in the venom duct. In this study, the disulfide connectivity and three-dimensional structure of these disulfide isomers were explored using regioselective synthesis, chromatographic coelution, and solution-state nuclear magnetic resonance spectroscopy. Of the native isomers, only the isomer with a ribbon disulfide configuration showed pharmacological activity similar to other χ-conotoxins. This isomer inhibited the rat norepinephrine transporter (IC_50_ = 10 ± 2 µM) and has the most structural similarity to previously characterized χ-conotoxins. In contrast, the globular isoform of PnID showed more than ten times less activity against this transporter and the beaded isoform did not display any measurable biological activity. This study is the first report of the pharmacological and structural characterization of an χ-conotoxin from a species other than *Conus marmoreus* and is the first report of the existence of natively-formed conotoxin isomers.

## 1. Introduction

Due to their ability to potently inhibit targets in pain pathways, venom peptides from the *Conus* genus are often studied for their pharmaceutical potential. Thousands of these peptides, known as conotoxins, have been characterized. The targets of these conotoxins encompass at least twelve different families of receptors [1,2], and it is usually the case that conotoxins bind their targets with a high degree of selectivity, which has allowed these peptides to be used as molecular probes [3]. The biological function of these cysteine-rich peptides is highly dependent on their tertiary structures which, similarly to most venom peptides, are stabilized by one or more disulfide bonds. The inter-cysteine loop size and cysteine framework of a conotoxin often correlates to its pharmacological target [4], so conotoxins with similar structures and molecular targets can be grouped into “pharmacological families”, each of which is denoted by a Greek letter. One such family, the χ-conotoxins, are known for their highly selective antagonism of the norepinephrine transporter (NET), making them potential therapeutic leads for diseases of the nervous system, including pain and epilepsy [5]. χ-Conotoxins contain a unique structural motif, characterized by a 4/2 inter-cysteine loop size (four residues included in the first intercysteine loop and two in the second) and the preferential formation of unique disulfide connectivity [6,7]. For conotoxins with four cysteines, three possible disulfide connectivity patterns are hypothetically achievable: the globular connectivity (pairings between C1-3 and C2-4), the ribbon connectivity (pairings between C1-4 and C2-3), and the beaded connectivity (pairings between C1-2 and C3-4). χ-Conotoxins are known for their preferential formation of a ribbon connectivity.

In this study, the venom profile of *Conus pennaceus* from the Red Sea was investigated using liquid chromatography with mass spectrometry (LC-MS), which led to the observation that the χ-conotoxin PnID exists in the venom as three native disulfide isoforms. Although PnID was previously identified by genetic analysis [4,8], this is the first report of the pharmacology, chemical synthesis and three-dimensional structure of this peptide. The peptide’s potential to form disulfide isomers was explored using regioselective solid-phase peptide synthesis and reverse-phase high-performance liquid chromatography (RP-HPLC), and confirmed by nuclear magnetic resonance (NMR) spectroscopy and conventional partial disulfide reduction and alkylation. This is the first report of the existence of native venom peptide isomers in *Conus* and the first report of a functional χ-conotoxin from a cone snail other than *Conus marmoreus.* PnID is also the first χ-conotoxin to be isolated from *Conus pennaceus*.

## 2. Results

### 2.1. Isolation and Identification of Native Isomers

LC-MS analysis (Figure 1A) showed that PnID is a highly abundant peptide within the venom duct of *Conus pennaceus* collected from the Red Sea (Figure 1C). Based on the previously identified sequence for PnID from genomic data [4,8], we expected the peak for the [M+2H]^2+^ ion to appear at *m*/*z* of 659.2 (for [M+2H]^2+^), which matched the observed *m*/*z* (Figure 1D). The extracted ion chromatogram for *m*/*z* = 659.2 ± 0.2 demonstrated the presence of three closely eluting chromatographic peaks (Figure 1A, blue trace). These peaks most likely correspond to the three possible disulfide bond isomers for PnID (see Section 2.2 and Section 2.3).

The peptide is also predicted to contain two disulfide bridges, which should result in an increase in *m*/*z* of two for the [M+2H]^2+^ ion after disulfide bond reduction (expected [M+2H]^2+^
*m*/*z* = 661.2). LC-MS analysis of the crude venom was carried out after disulfide bond reduction (Figure 1B) by tris-carboxyethyl phosphine (TCEP). The extracted ion chromatogram (Figure 1B, blue) corresponding to the [M+2H]^2+^ charge state of the fully reduced peptide (*m*/*z* = 661.2 ± 0.2), yielded a single peak with an observed *m*/*z* value of 661.3 that was very close to the expected value (Figure 1E). The broadening of the range for the extracted ion chromatogram did not reveal the presence of any additional chromatographic peaks nor were any peaks seen for an extracted ion chromatogram for *m*/*z* = 659.2 ± 0.2 for this reduced sample (data not shown). This evidence led to the conclusion that three native disulfide isomers were present for a single conotoxin from the venom duct of *C. pennaceus*.

Reduced PnID was isolated by RP-HPLC and subjected to Edman degradation, which confirmed the primary sequence. PnID was then chemically synthesized using Fmoc solid-phase peptide synthesis and randomly oxidized to form disulfide bonds. After oxidation, RP-HPLC of the synthetically produced peptide showed two nearly equal peaks, both of which matched the expected mass of PnID and had relative retention times similar to the latter two peaks present in the native venom (Figure 2A).

### 2.2. Disulfide Bond Determination by HPLC Co-Elution Experiments

To independently confirm the disulfide bond connectivity of PnID, the sequence was subjected to both single-step ‘random’ oxidations and two-step regioselective oxidations corresponding to each of the three specific disulfide bond pairings achievable [PnID A (globular), PnID B (ribbon), and PnID C (beaded), Figure 2]. Interestingly, the single step oxidation appeared to result in the production of only two major disulfide isomers with masses that corresponded to oxidized PnID, unlike the native material which appeared to have three isomers. RP-HPLC co-elution was attempted for mixtures of random and regioselective products. The experiments were carried out with the isomers at equal concentrations and roughly at a ratio of 1:2 (peptides produced by single-step oxidation and two-step oxidation, respectively). PnID A appeared to completely co-elute with the earlier eluting major peak from the single-step oxidation and displayed no distinct separation or shouldering (retention time, *t*_R_, of ~31 min; Figure 2B). PnID B was observed to co-elute with the later peak in the air oxidation profile. The result was one sharp peak without evidence of shouldering and with a retention time (*t*_R_) of ~34 min (Figure 2C). PnID C (beaded isoform), which contained disulfide bonds between cysteines 3–4 and 9–12 did not appear to co-elute with any of the peaks generated from the single-step oxidation as it generated a large shoulder peak (*t*_R_ of ~33 min; Figure 2D). It is possible that a small amount of C is generated in the random oxidation but is too unfavorable to be seen as a main peak and is instead eclipsed by the PnID B peak. These results, in combination with the PSD-MALDI-MS data, confirm the assignment of a globular disulfide bond configuration to the earlier synthetic peak (PnID A) and a ribbon configuration to (PnID B).

### 2.3. Disulfide Bond Determination by Reduction and Alkylation

To further confirm the disulfide bond connectivity of the air oxidized material, conventional disulfide bond reduction and alkylation was undertaken. This analysis was performed using N-phenyl maleimide and 4-vinyl-pyridine as differential alkylating agents, followed by post-source decay—matrix-assisted laser desorption/ionization—mass spectrometry (PSD-MALDI-MS) sequence assignment, to identify individual alkylation attachments. For peptides containing four cysteines, three specific disulfide pairings known as globular (Cys I-III, Cys II-IV), ribbon (Cys I-IV, Cys II-III), and “beads on a string” or beaded forms (Cys I-II, Cys III-IV), are hypothetically achievable (Figure 2B–D). Based on the MS data, PnID A was assigned a globular configuration with 3–9 and 4–12 disulfide linkages (Appendix A). PnID B presented several challenges due to the rapid opening of both disulfide bonds under a controlled TCEP reduction (Appendix A). A four-fold decrease in TCEP was required to selectively open the first disulfide bond in PnID B in comparison to PnID A. The 3–12 disulfide linkage (corresponding to the ribbon isoform) was identified in PnID B with certainty and the 4–9 disulfide linkage was inferred. No PSD ions were observed for the predicted PnID beaded isomer (disulfide linkage 3–4 and 9–12). The fully oxidized material was also subjected to PSD analysis which, if beaded, should have resulted in significant peptide fragmentation, but this was not the case.

### 2.4. Three-Dimensional Structures of PnID

The NMR-derived structures for the globular and ribbon forms of PnID are shown in Figure 3. Whereas the globular form of PnID (PnID A) does not exhibit any regular secondary structure elements, the ribbon form (PnID B) contains a β hairpin, similar to what has been observed in other χ-conotoxins. Although not apparent from the ensembles, the broader lines in the NMR spectra for PnID B (data not shown) indicate increased dynamics relative to PnID A, a difference that we hypothesize arises from the globular form of PnID being more tightly packed (and therefore more rigid) than the ribbon form. Interestingly, the ribbon form of PnID (PnID B) shows a high degree of structural homology to the χ-conopeptide MrIA (Figure 4), which has also been shown to target the neuronal noradrenaline transporter [9]. The degree of structural similarity between these peptides is perhaps unsurprising based on the degree of homology in their primary sequences and their identical cystine bond topologies (I-IV, II-III).

### 2.5. Pharmacology of the PnID Isomers

Due to the high sequence and structural similarity of PnID to the χ-conotoxins, experiments were conducted to see if it was indeed a member of this pharmacological family. To do this, the ability of the PnID isomers to inhibit the uptake of tritium-labeled neurotransmitters ([^3^H]serotonin, [^3^H]norepinephrine, and [^3^H]dopamine) by their respective monoamine transporters [the serotonin transporter (SERT), the norepinephrine transporter (NET), and the dopamine transporter (DAT)] was examined in comparison to the effects of the nonselective inhibitor cocaine. The data in Figure 5 demonstrate that the most potent and selective isomer of PnID is the ribbon isoform (PnID B), which is the disulfide configuration that all previously described χ-conotoxins natively adopt [10]. In particular, PnID B selectively inhibited uptake at NET (IC_50_ = 10 ± 1.75 μM). In contrast, PnID A and C were determined to be ineffective at inhibiting uptake at these transporters, with estimated IC_50_ values in excess of 100 μM for all three transporters.

## 3. Discussion

This study aimed to investigate the activity and structure of a novel conotoxin isolated from *Conus pennaceus*. At the beginning of this project, we determined that several native isomers of the novel peptide, PnID, were present within the venom duct. These observations led to the work to determine how these isomers differed in their ability to inhibit their target(s), one of which we identified as the NET.

Interestingly, when we attempted to synthetically reproduce the three disulfide isomers seen in the native venom using a non-selective air oxidation strategy, we were only able to produce two isomers, despite three isomers being theoretically possible and three isomers appearing in the native venom. We determined that the isomers from the single-step air oxidation were the globular and ribbon forms of PnID. Although it cannot be directly stated what the three isomers in the native venom are, it is plausible that they correlate to the respective ribbon, globular, and beaded forms since they were all reduced by TCEP. The native presence of a beaded isomer would be unusual as this isomer contains energetically costly vicinal disulfide bonds. However, despite the higher energy cost of producing native vicinal disulfides, such bonds have been reported previously in nature albeit rarely [11]. An additional possibility is that the isomer that was unable to be produced in the single-step oxidation is a dimer, instead of an intramolecular disulfide isomer. It is also plausible that the beaded isomer is too energetically unfavorable to be produced in the single-step oxidation, whereas the environment of the venom duct allows for its favorable production due to the presence of chaperone proteins. Other possibilities to the identity of this third isomer could be the inclusion of additional dihedral angles of the cysteine residues or the cis isomerization of proline, either of which may be favored in the native state due to the presence of chaperones. Despite this, the globular and beaded isoforms appeared to be much less active in inhibiting the rat NET, which fits the pharmacophore model previously proposed for χ-conotoxins. In this model, it has been well documented that globular isoforms of the known χ-conotoxins were far less potent than their ribbon form [12]. Previously studied conotoxins have always been characterized as having only a single active native isomer. Deviations from the disulfide connections formed in natural conotoxins, such as those found in synthetic disulfide isomers, often display reduced potency. For example, the χ-conotoxin CmrVIA and several α-conotoxins (nAChR antagonists) have, for the most part, been significantly less active (a 10-fold reduction or more) than their native counterparts [12,13].

However, rare exceptions to these observations have been reported in which non-native isoforms retain activity or even have increased activity compared to the native isoform. One example is the α-conotoxin AuIB, in which the “non-native” ribbon isoform has a 10-fold higher activity than the native globular form [14]. This phenomenon has also been reported in the µ-conotoxin family (Na_v_ channel inhibitors) where multiple non-native isoforms of the µ-conotoxin KIIIA displayed high levels of potency [15].

Characterizing native disulfide bond connectivity within peptide toxins is often difficult, requiring large quantities of peptide and multiple experiments [16]. As disulfide bond formation is a post-translational modification that cannot be easily predicted on a genetic level, these observations of greater activity for “non-native” isoforms hint that disulfide bond rearrangement may actually be a previously undiscovered venom diversification strategy for *Conus*, which could add to their already extensive repertoire of post-translational modification strategies. Our discovery of three disulfide isomers within the native venom of *C. pennaceus* lends credence to this hypothesis.

The sequence of PnID, although relatively homologous to other χ-conotoxins (Table 1), is unique in its N- and C-terminal regions. As previously mentioned, a well-established pharmacophore has been developed for χ-conotoxins; however, much of the work conducted to establish this model was performed with single residue replacements [10]. In contrast, recent work has highlighted the importance of multiple residue replacements and their influence on structure-activity relationships [17]. Considering the disparity in the N- and C-terminal regions of PnID with previously studied χ-conotoxins, all of which are truncations or modifications of χ-MrIA, an investigation of the contribution of these regions to the overall potency and selectivity of the peptide through the generation of chimeric peptides may be a worthwhile endeavor.

Structurally, the ribbon isomer of PnID appears to be fairly similar to that of the known χ-conotoxins, with its first intercysteine loop appearing less rigid than the rest of the molecule. This does not appear to be the case for the more compact globular form, which appeared to be more structurally stable based on the NMR data.

It is worth noting that a limitation of this study is our use of rat transporters for detecting PnID activity. Xen2174, a highly potent synthetic analog of χ-MrIA tested in rat synaptosomes, previously entered a phase II clinical trial, where it failed to meet its endpoint [19]. This failure has since been attributed to its lack of potency against the human NET due to structural differences between the two transporters. Therefore, testing against human transporters would offer a clearer perspective on PnID’s true potential as a pain therapeutic.

Finally, all previously known χ-conotoxins have been isolated from *Conus marmoreus*. PnID is the first χ-conotoxin to be isolated from a separate species. A recent article by Ziegman et al. demonstrated that messy processing of the χ-MrIA template leads to various active homologs of this toxin within *C. marmoreus* [18]. If a similar type of processing is used by *C. pennaceus*, this could indicate that PnID may be the first of several leads for targeting the NET that can be isolated from this organism.

## 4. Materials and Methods

### 4.1. Venom Duct Extraction

Several specimens of *C. pennaceus* were collected from the Red Sea. The specimens were sacrificed, and whole venom ducts were dissected and prepared as previously described [20] (essentially, the venom ducts were homogenized into a fine powder and weighed). Peptide extraction was achieved with 95% Solvent A (0.1% *v*/*v* TFA/H_2_O) and 5% Solvent B (90/10/0.008% *v*/*v* CH_3_CN/H_2_O/TFA), with samples typically at a concentration of approximately 1 mg mL^−1^. Samples were vortexed and then sonicated for 10 min. Extracts were then centrifuged at 12,000× *g* for 10 min. The resulting supernatants were removed, dried using a vacuum concentrator, weighed, and then stored at −20 °C until required. All extracts were resuspended in the above solvent at 1 mg mL^−1^, sonicated (5 min), and re-centrifuged (at 12,000× *g* for 10 min) prior to chromatographic separation and MS analysis.

### 4.2. Chromatographic Separation

Native and synthetic conotoxins were individually separated using the following equipment: (i) capillary scale (Phenomenex; C18, 5 μm, 300 Å, 1.0 × 250 mm, flow 100 μL min^−1^)—used for comparative RP-HPLC profiling to control the quality of peptide purity, to quantify the peptides and to perform peptide co-elution experiments, (ii) analytical scale (Vydac; C18, 5 μm, 300 Å, 4.2 × 250 mm, flow 1 mL min^−1^)—used for the isolation and purification of native peptides for MS analysis, and (iii) preparative scale (Vydac; C18, 10 μm, 300 Å, 22 × 250 mm, flow 5 mL min^−1^)—used for the preparative separation of synthetic peptides. Capillary and analytical scale separations utilized a Waters 2695 Alliance RP-HPLC System interfaced with a 996 Waters photo diode array detector. Data was acquired and analyzed using Waters Millennium32 (v3.2) software. Samples were eluted using a linear 1% min^−1^ gradient of organic (90/10%/0.008% *v*/*v* CH_3_CN/H_2_O/TFA) Solvent B against aqueous (0.1% *v*/*v* TFA/H_2_O) Solvent A for 65 min, terminating with a high organic wash (80% Solvent B for 5 min), and pre-equilibration step (5% Solvent B) for 10 min prior to sample injection. Elution from the column was monitored at 214 nm. The preparative scale RP-HPLC (iii) used a 625 Waters HPLC pump and controller interfaced with a 996 Waters photo diode array detector. Synthetic peptides and crude venom peptide extracts were filtered (Nylon 0.22 μm), manually loaded, and eluted from the preparative scale column using the same 1% gradient at 5 mL min^−1^ and monitored at 214 and 280 nm. Fractions were collected manually and stored at −20 °C or freeze-dried until required.

LC-MS analysis of crude duct venom was performed on a C18 capillary-bore RP-HPLC (Phenomenex; 5 μm, 300 Å, 1.0 × 250 mm) column interfaced to a PerSeptive Biosystems Mariner MS, using a 1% gradient, 100 μL min^−1^, 214 nm, Solvents (A) 0.1% formic acid/H_2_O, (B) 0.65% formic acid/CH3CN, with only 20% of the flow directed into the MS ion source. To achieve total venom peptide reduction, materials were resuspended in 200 mM TCEP/25 mM NH_4_OAc, pH 4.5, heated at 60 °C for 10 min, then centrifuged (12,000× *g*) prior to LC-MS analysis.

### 4.3. Disulfide Bond Connectivity Analysis by Reduction and Alkylation

Disulfide connectivity analysis was achieved in real-time by ESI-MS direct infusion as previously described [21]. Alkylation reactions were sequentially carried out on partial and fully reduced forms using N-phenyl maleimide (100 mM in isopropanol) and freshly distilled 4-vinyl-pyridine, respectively. RP-HPLC purified material, partially reduced and alkylated, and fully reduced-differentially alkylated, were subjected to both PSD-MALDI-MS analysis and Edman degradation.

Reduced non-alkylated and alkylated RP-HPLC purified peptide derivatives (~40–250 pmoles) were applied to polybrene-treated glass fiber support filters for automated Edman degradation on a gas-phase sequencer (model 470A; Applied Biosystems, Foster City, CA, USA). Identification of sequential amino acids followed the method described by Atherton et al. [22] and Matsudaira [23]. Identified sequence information was calculated and confirmed with corresponding MS data. Alkylated positions were identified as a unique PTH-N-phenyl maleimide doublet or as a pyridylethylated cysteine standard.

### 4.4. Synthetic Peptide Production

Conotoxins were manually assembled on 4-(methyl)benzhydrylamine (MBHA) Rink Amide resin (0.44 meq g^−1^) using 9-fluorenylmethoxycarbonyl (Fmoc) chemistry as previously described [24]. Side chain protecting groups included: Ser(tBu), Thr(tBu), Cys(Trt), Arg(Pbf), and Tyr(tBu), with the addition of Cys(acm) on positions 3 and 12 on isomer PnID B (ribbon 1), positions 3 and 9 on isomer PnID A (globular 1), and positions 9 and 12 on isomer PnID C (beaded). Upon completion of synthesis, the peptidyl-resin was washed with DMF (2x, 5 mL) followed by dichloromethane (DCM; 10 mL) and dried under N_2_.

Assembled peptides were cleaved and oxidized as previously described [24]. Assembled peptides were cleaved using Reagent K [TFA (82.5% *v*/*v*), phenol (5% *v*/*v*), water (5% *v*/*v*), thioanisole (5% *v*/*v*), and triisopropylsilane (TIPS) (2.5% *v*/*v*)]. Forty (40) mL of Reagent K per gram of peptidyl-resin was mixed for 2 h at 24 °C after which the cleaved slurry was vacuum filtered into liquid N_2_ chilled tert-butyl methyl ether. The peptide precipitate was pelleted by centrifugation (3000× *g*, 10 min) and washed twice with chilled tert-butyl methyl ether. The resulting peptide pellet was suspended in 25% *v*/*v* acetic acid, then freeze-dried to form a powder and stored at −20 °C until required. Crude peptides (1 mg mL^−1^) were oxidized using 100 mM NH_4_HCO_3_, pH 8 and stirred for 5 days at 4 °C. Oxidized material was filtered (0.45 μm) prior to semi-preparative RP-HPLC fractionation.

Cleaved χ-conotoxin isomers PnID A, PnID B, and PnID C, were RP-HPLC purified and then oxidized, as above. As confirmed by ESI-MS, partially oxidized materials were subjected to spontaneous thiol deprotection and disulfide bond formation as previously described [24]. Deprotection was achieved by dissolving the partially folded peptide in 50% *v*/*v* acetic acid/H_2_O (1 mg mL^−1^) and by adding a solution of freshly saturated I_2_ in 50% *v*/*v* acetic acid/H_2_O to the stirring peptide (25% reaction vol.). The reactions were quenched after 5 min with the addition of 10 μL aliquots of 1 M Na_2_S_2_O_3_ until the stirring solution became clear, which was then followed by the addition of 200 μL of 0.1% TFA/H_2_O. The resulting acidified material was centrifuged (12,000× *g*, 5 min) and directly purified by preparative RP-HPLC (as above) with mass confirmation provided by ESI-MS.

### 4.5. Disulfide Bond Connectivity Analysis by HPLC Co-Elution

Each of the PnID peptide isomers, as well as the products from air oxidation, were co-eluted both in a 1:1 and 2:1 ratio on a C18 capillary-bore RP-HPLC (Phenomenex; 5 μm, 300 Å, 1.0 × 250 mm) column using the methods and equipment as described in the chromatographic separation and analysis section (see above).

For confirmation of the production of our synthetic peptides, an AB/MDS-Sciex API 3000 triple quadrupole mass spectrometer (Thornhill, ON, Canada) was used in this investigation as previously described [20]. The ESI-MS system was calibrated manually in positive mode with PPG 3000 (AB/MDS-Sciex) to achieve <5 ppm mass accuracy per the manufacturer’s protocol.

### 4.6. Animal Care

Male Sprague-Dawley rats (Envigo, Frederick, MD, USA) weighing 250–350 g were housed three per cage with ad lib access to food and water and maintained on a 12 h light/dark cycle. Animal facilities were accredited by the Association for the Assessment and Accreditation of Laboratory Animal Care, and procedures were carried out in accordance with the Institutional Animal Care and Use Committee and the National Institutes of Health guidelines on the care and use of animal subjects in research.

### 4.7. Pharmacology

[^3^H]Dopamine, [^3^H]norepinephrine, and [^3^H]5-HT (specific activity ranging from 30–50 Ci/mmol) were purchased from Perkin Elmer (Shelton, CT, USA). All other chemicals and reagents were acquired from Sigma-Aldrich (St. Louis, MO, USA). Rats were euthanized by CO_2_ inhalation, and brains were processed to yield synaptosomes as previously described [25]. Rat caudate tissue was used for DAT assays, whereas rat whole brain minus caudate and cerebellum was used for NET and SERT assays. Transport activity at DAT, NET, and SERT was assessed using 5 nM [^3^H]dopamine, 10 nM [^3^H]norepinephrine, and 5 nM [^3^H]5-HT, respectively. The selectivity of uptake assays was optimized for a single transporter by including unlabeled blockers to prevent uptake of [^3^H]transmitter by competing transporters. Uptake inhibition assays were initiated by adding 100 µL of tissue suspension to 900 µL Krebs-phosphate buffer (126 mM NaCl, 2.4 mM KCl, 0.83 mM CaCl_2_, 0.8 mM MgCl_2_, 0.5 mM KH_2_PO_4_, 0.5 mM Na_2_SO_4_, 11.1 mM glucose, 0.05 mM pargyline, 1 mg mL^−1^ bovine serum albumin, and 1 mg mL^−1^ ascorbic acid, pH 7.4) containing test peptide and [^3^H]transmitter. Assays were terminated by rapid vacuum filtration through Whatman GF/B filters, and retained radioactivity was quantified by liquid scintillation counting. Statistical analyses were carried out using GraphPad Prism 6.0 (GraphPad Scientific, San Diego, CA, USA). IC_50_ values for uptake inhibition were calculated based on non-linear regression analysis.

### 4.8. NMR Measurements and Analysis

For NMR analysis, lyophilized peptides were resuspended at a concentration of 2 mM (PnID A) or 1 mM (PnID B) in 95% H_2_O/5% D_2_O with 0.2 mM 2,2-dimethylsilapentane-5-sulfonic acid (DSS) as a chemical shift reference. The samples were adjusted with HCl to pH 4.0.

All NMR spectra were collected with a sample temperature of 5 °C using a Bruker (Billerica, MA) Avance NEO 600 MHz spectrometer with a room-temperature TXI probe. Proton chemical shift assignments were made using ^1^H-^1^H TOCSY (60 ms mixing time) and ^1^H-^1^H NOESY spectra (100 ms mixing time) experiments. Carbon and nitrogen resonances were assigned using the natural abundance signals measured by ^1^H-^13^C HSQC and ^1^H-^15^N BEST-HSQC experiments. NOE-based distance constraints were generated from ^1^H-^1^H NOESY spectra (600 ms mixing time). ^3^*J*_HN-Hα_ values were measured directly from 1D ^1^H spectra. NMR data were processed using TopSpin 4.1.4 (Bruker) and assigned using CCPN Analysis 2.5.2 [26].

### 4.9. Structure Calculation

Structures were initially calculated with ARIA 2.3 [27], using chemical shift-matched NOESY peaks lists, backbone dihedral angle restraints (φ, ψ) generated using TALOS-N [28], ^3^*J*_HN-Hα_ couplings, and the known cystine topology as inputs. Later rounds of structure calculation included hydrogen-bond restraints determined from the preliminary structures and side-chain dihedral constraints (χ_1_, χ_2_) for cystines determined using DISH [29]. Only DISH predictions with confidence above 80% were used. The standard ARIA protocol was used except that 30 structures were calculated in each iteration (other than for iteration 8, in which 100 structures were calculated), the log-harmonic potential [30] was enabled, the spin diffusion correction was enabled, and the number of simulated annealing steps was tripled in the two cooling periods (cool1 and cool2). The 20 structures with the lowest energy from iteration 8 were selected for water refinement and used to generate the structural ensemble. Statistics describing the restraints used for the calculations and the resulting ensemble are provided in Table 2. Chemical shifts and restraints were deposited at the Biological Magnetic Resonance Bank [BMRB: 30991 (PnID A) and 30992 (PnID B)], and atomic coordinates were deposited in the Worldwide Protein Data Bank [PDB: 7TVQ (PnID A) and 7TVR (PnID B)].

## 5. Conclusions

We have reported for the first time the structure and pharmacology of a χ-conotoxin that originates from a species other than *C. marmoreus*. Additionally, we have found that this toxin is produced as native isomers. The ribbon isoform shows much greater activity against the NET than the other isomers, which hints at a currently unknown function of these isomers. This work also proves that molluscivores outside of *C. marmoreus* can produce χ-conotoxins, thus expanding the potential for identifying novel leads for modulators of neuronal proteins, as this family of conotoxins likely exists in other species as well.

## Figures and Tables

**Figure 1 marinedrugs-21-00061-f001:**
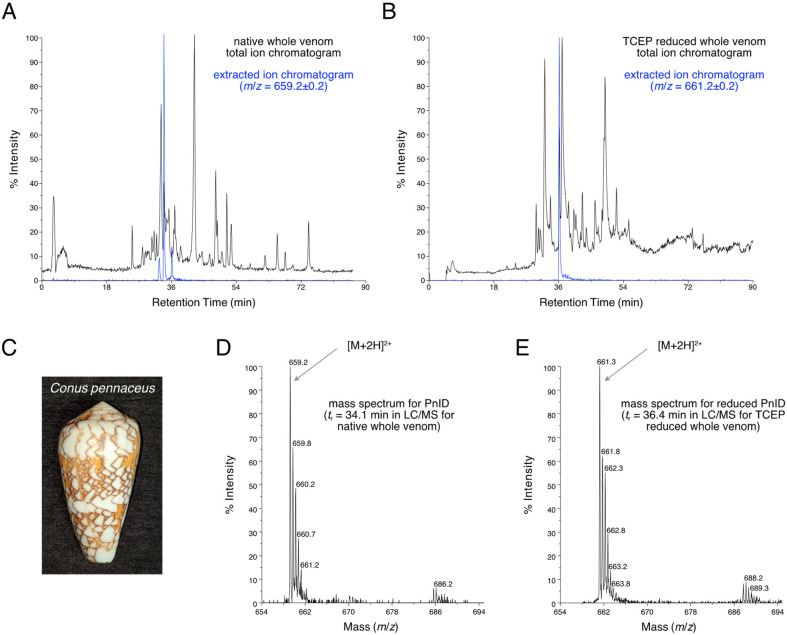
(**A**). LC-MS total ion chromatogram of the crude duct venom extract of Red Sea *C. pennaceus* (pictured in (**C**)). The extracted ion chromatogram for *m*/*z* = 659.2 (shown in blue) has peaks corresponding to the [M+2H]^2+^ ions of the native PnID peptides. (**B**). The total ion chromatogram LC-MS after TCEP reduction. The extracted ion chromatogram for *m*/*z* = 661.2 (shown in blue) has only a single peak, corresponding to the [M+2H]^2+^ ion of fully reduced PnID. (**D**). Mass spectrum corresponding to a retention time of 34.1 min in the LC-MS data shown in (**A**). has an *m*/*z* value consistent with the expected value for the [M+2H]^2+^ ion of fully oxidized PnID. (**E**). Mass spectrum corresponding to a retention time of 36.4 min in the LC-MS data shown in (**B**). has an *m*/*z* value consistent with the expected value for the fully-reduced [M+2H]^2+^ ion of PnID.

**Figure 2 marinedrugs-21-00061-f002:**
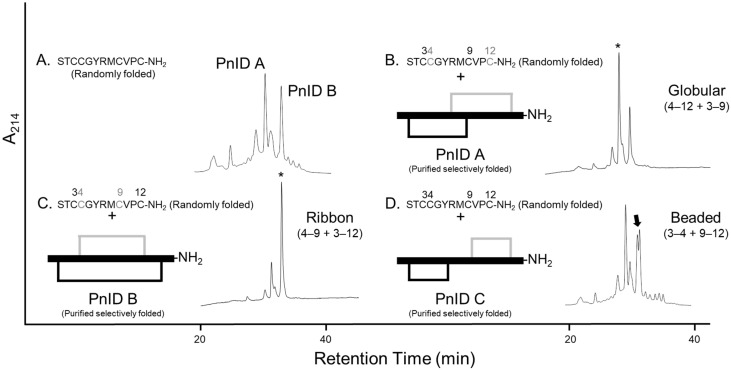
Comparative RP-HPLC analysis of the single-step (random) oxidized isomers of PnID to the regioselectively synthesized forms. Numbers above each sequence indicate the positions of the cysteines. (**A**). The isomeric products of the random air oxidation of PnID. (**B**–**D**). Co-elution of the selectively folded material with the random air oxidation products. * Indicates position of the mixed peak, the arrow indicates the appearance of shouldering and a lack of coelution. (**B**,**C**). are the same scale, (**A**,**D**). are rescaled to fit.

**Figure 3 marinedrugs-21-00061-f003:**
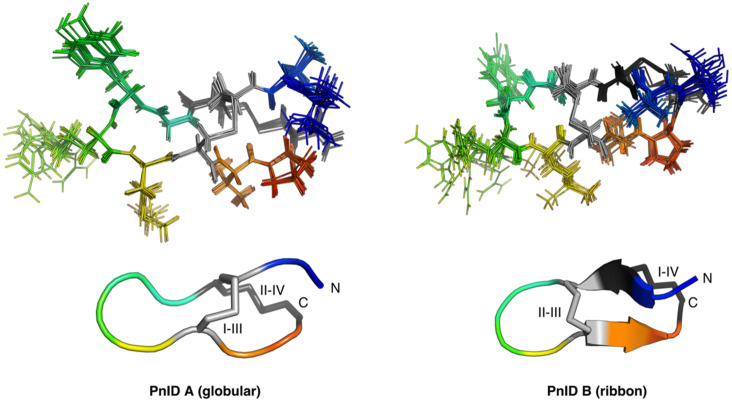
Ensemble (**top**) and cartoon (**bottom**) representations of PnID A (**left**) and PnID B (**right**). The 20 lowest-energy structures from 100 structures calculated in the final iteration of the ARIA protocol were refined in explicit water and used to generate the structural ensembles. The N-termini are colored blue, the C-termini are colored red, and the disulfide bond topologies are indicated by Roman numerals. The same orientation and scale are used for the ensemble (**top**) and cartoon (**bottom**) representations. The cartoon representations show the lowest energy structure from each ensemble. For purposes of clarity, only 10 of the 20 structures used to define the structural ensemble are shown here with residues 2–11 aligned.

**Figure 4 marinedrugs-21-00061-f004:**
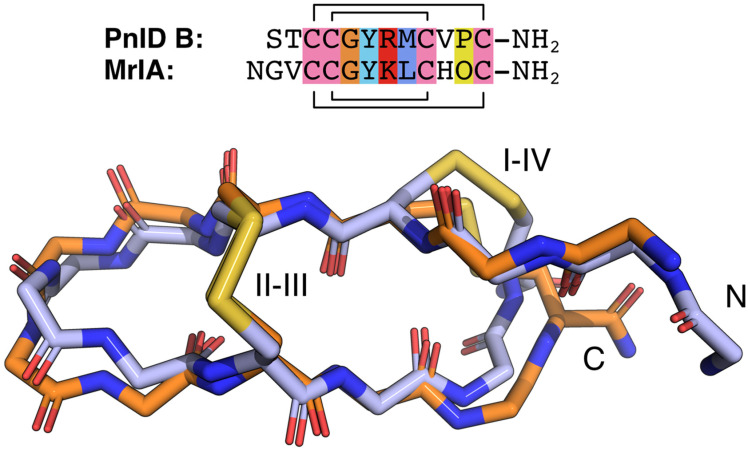
Sequence and backbone alignments for PnID B (orange) and MrIA (blue). The backbone atoms for the lowest energy structure for the PnID B structural ensemble (in orange) were aligned against the backbone atoms for the lowest energy structure (in blue) from the ensemble for MrIA (PDB ID 2EW4). The sequence alignment for these peptides is provided above and colored following the ClustalX default coloring scheme. In the MrIA sequence, “O” represents hydroxyproline. Both peptides have an amidated C-terminus (NH_2_).

**Figure 5 marinedrugs-21-00061-f005:**
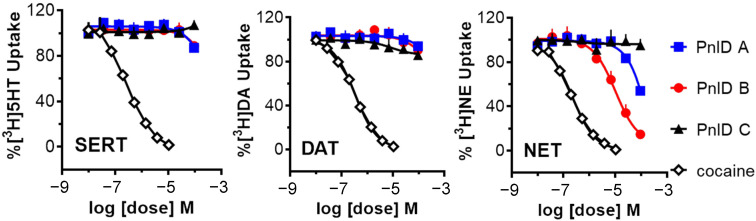
Uptake inhibition plots showing the comparative dose-response inhibition of the monoamine transporters SERT (**left**), DAT (**middle**), and NET (**right**) by PnID A, PnID B, PnID C, and cocaine. Whereas the PnID isomers do not show inhibition at SERT and DAT, the ribbon isoform (PnID B) shows activity for inhibiting NET. Statistical analyses were carried out using GraphPad Prism 6.0 (GraphPad Scientific, San Diego, CA, USA). IC_50_ values for uptake inhibition were calculated based on non-linear regression analysis.

**Table 1 marinedrugs-21-00061-t001:** Comparison between the mature regions, activity and folding of the reported χ-conotoxins from the T-superfamily.

χ-Conotoxin	Mature Sequence	IC_50_ NET	Cystine Connectivity	Ref.
PnID (Ribbon form)	STCCGYRMCVPC *	10 μM	1–4 & 2–3	This work
MrIA	NGVCCGYKLCHOC *	645 nM	1–4 & 2–3	[9,12,18]
MrIB	VGVCCGYKLCHOC *	860 nM	1–4 & 2–3	[7]
CmrVIA	VCCGYKLCHOC *	N/A	1–4 & 2–3	[12]

* C-terminal amidation; O, 4-trans-hydroxyproline; green highlight, sequence identity; yellow highlight, sequence homology. Comparing PnID to MrIA: six of 13 residues are identical (46.1%, including all cysteines) and another three residues are homologous (23.1%).

**Table 2 marinedrugs-21-00061-t002:** Structural statistics for the PnID A and PnID B ensembles.

			PnID A	PnID B
Physical parameters				
	Number of residues		12	12
	Average molecular weight (unlabeled, Da)		1317.6	1317.6
Structural restraints				
	NOE-derived distance restraints (ARIA cycle 8)			
		Intraresidue (| *I* − *j* | = 0)	90	72
		Sequential (| *i* − *j* | = 1)	85	42
		Short (2 ≤ | *i* − *j* | ≤ 3)	13	7
		Medium (4 ≤ | *i* − *j* | ≤ 5)	4	1
		Long (| *i* − *j* | > 5)	13	8
		Ambiguous	29	15
		Total	234	145
	Chemical shift-based dihedral constraints			
		*φ* (from TALOS-N)	6	5
		*ψ* (from TALOS-N)	6	5
		Cystine *χ*_1_ and *χ*_2_ (from DISH)	3	5
	Scalar coupling backbone torsion restraints (^3^*J*_HNHA_)		9	0
	Hydrogen-bond restraints		1	4
	Disulfide bond restraints		2	2
Statistics for accepted structures				
	Accepted structures		20 of 100	20 of 100
	Mean CNS energy terms			
		*E* total (kcal mol^−1^)	−324 (±9)	−341(±12)
		*E* van der Waals (kcal mol^−1^)	−33.3 (±1.2)	−32.3 (±1.8)
		*E* distance restraints (kcal mol^−1^)	70.03 (±0.06)	44.013 (±0.002)
	Restraint violations (average # per structure)			
		NOE (>0.5 Å)	0.4 (±0.6)	2.2 (±1.2)
		Dihedral (>5°)	0	0
		^3^*J*_HNHA_ (>1 Hz)	1.1 (±0.2)	N/A
	RMS deviations from the ideal geometry used within CNS			
		Bond lengths (Å)	3.25 × 10^−3^ (±1.5 × 10^−4^)	3.13 × 10^−3^ (±1.7 × 10^−4^)
		Bond angles (°)	0.41 (±0.02)	0.36 (±0.02)
		Improper angles (°)	1.3 (±0.2)	1.3 (±0.2)
		Dihedral angles (°)	39.8 (±0.3)	41.0 (±0.7)
Ramachandran statistics (PROCHECK 3.5.4, [31])				
	Most favored (%)		99.4 (±2.8)	86.9 (±2.8)
	Additionally allowed (%)		0.6 (±2.8)	10.6 (±4.6)
	Generously allowed (%)		0	2.5 (±5.1)
	Disallowed (%)		0	0
MolProbity analyses (v3.19, [32])				
	Clashscore		1.2 (±2.4)	4.7 (±3.1)
	Clashscore percentile (%)		98.2 (±3.7)	91.9 (±7.7)
Average atomic RMS deviations from average structure (±SD)				
N, C_α_, C, and O atoms (all residues, Å)			0.23 (±0.11)	0.38 (±0.09)
All heavy atoms (all residues, Å)			0.60 (±0.17)	1.02 (±0.28)
N, C_α_, C, and O atoms (for residues 2-11, Å)			0.17 (±0.06)	0.30 (±0.10)
All heavy atoms (for residues 2-11, Å)			0.60 (±0.17)	1.02 (±0.30)

## Data Availability

NMR Chemical shifts and restraints were deposited at the Biological Magnetic Resonance Bank [BMRB: 30991 (PnID A) and 30992 (PnID B)], and atomic coordinates were deposited in the Worldwide Protein Data Bank [PDB: 7TVQ (PnID A) and 7TVR (PnID B)].

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
