# Peer review of "Characterization of the Native Disulfide Isomers of the Novel χ-Conotoxin PnID: Implications for Further Increasing Conotoxin Diversity"

_marinedrugs, 2023, doi:10.3390/md21020061_

Round 1

Reviewer 1 Report

In this manuscript, the authors identified three native disulfide isomers of χ-Conotoxin PnID from the venom of Conus pennaceus. The three disulfide isomers were synthesized using a regioselective folding strategy. The structure of the ribbon isomer was elucidated. This isomer still inhibited the rat norepinephrine transporter with low affinity. Here, I find some fundamental questions remain to be addressed and more supporting experiments are needed.

1. The name of Conus should be italicized, such as in line 65.

2. In Figure1A and B, the authors provide the data of native and reduced ion chromatograms. The extracted ion peaks in figure A are separated. In this study, authors should purify three isomers from the venom, and show their coelution peaks using analytical RP-HPLC. Meanwhile, the native and reduced molecular weights of three isomers are monitored by MS.

3. In Line 88, Fig. 1C should be revised.

4. PnID isomers were chemically synthesized using the random and regioselective folding methods. In figures 2 B, C and D, randomly folded should be modified as selectively folded. To my understanding, no more than three isomers are formed after random air folding with four cysteines. Why are there so many gradients after random oxidation in figure 2A?

5. In figure 2C, co-elution of the selectively folded material with the random air oxidation. However, the background peaks do not look like the result of random air oxidation.

6. To convince me, authors should coelute three synthetic isomers with three native PnID from venom.

7. In section 2.3, the authors identified the disulfide bond of synthetic PnID by reduction and alkylation. More importantly, authors should determine the disulfide bond linkage of native PnID purified from venom.

8. The annotations of Figure 3 and Figure 4 mix up.

9. Statistical analysis methods are required for pharmacological activity in Figure 5.

10.  Authors should modify and improve the quality of the Figures in Supplementary Materials.

11.  In the disulfide bond determination by reduction and alkylation section (line134), the authors identified sequence using matrix-assisted laser desorption/ionization mass spectrometry (PSD-MALDI-MS), but I do not see any description of this experiment in Materials and Methods section. Moreover, the authors identified the secondary ion fragments using tandem mass spectrometry (Supplemental Fig. 3), the authors did not analyze the data clearly.

Author Response

Point 1. The name of Conus should be italicized, such as in line 65.

Response 1. Thank you for catching this. Conus on line 65 is now italicized.

Point 2. In Figure1A and B, the authors provide the data of native and reduced ion chromatograms. The extracted ion peaks in figure A are separated. In this study, authors should purify three isomers from the venom, and show their coelution peaks using analytical RP-HPLC. Meanwhile, the native and reduced molecular weights of three isomers are monitored by MS.

Response 2. Thank you for this comment. While we agree that such an experiment would be ideal, the native venom was originally isolated from Conus pennaceus from the Red Sea. The native peaks were also identified a number of years ago and new venom would need to be collected. Such a task would require a larger amount of funding and resources than we have at present and would make it exceedingly difficult to complete this project. Additionally, purifying the native isomers would be fairly difficult as the amount of peptide available in native venom is low. The peptides would also need to be isolated in their oxidized form to retain their disulfide bond configurations. This would likely require many purification attempts on an already small amount of peptide, further increasing the difficulty of this task. However, all of the raw data from these experiments are available if the reviewer would like to view the ion extractions themselves.

Point 3. In Line 88, Fig. 1C should be revised.

Response 3. This has been removed.

Point 4. PnID isomers were chemically synthesized using the random and regioselective folding methods. In figures 2 B, C and D, randomly folded should be modified as selectively folded. To my understanding, no more than three isomers are formed after random air folding with four cysteines. Why are there so many gradients after random oxidation in figure 2A?

Response 4. Panels B, C, and D contain both randomly and selectively folded material (i.e., these are standard addition experiments). The sequence at the top of the frame labeled randomly folded indicates that this peptide was mixed with the selectively folded material at the bottom of each frame. Since Panel A does not contain any selectively folded material, this is not present in the description. The intent of the plus sign in panels B, C, and D is to convey that these chromatograms are of mixtures. If you have suggestions for how to adapt this figure to be more straightforward we can incorporate them.

You are correct in that only three isomers should be theoretically achievable due to disulfide connections. However, as was pointed out by other reviewers it is possible that isomerization of proline from trans to cis could be contributing to the formation of additional products in the randomly folded material. It may also be possible that the cysteines are adopting additional dihedral angles that contribute to the formation of different products (Daly and Craik, IUBMB Life 2009 'Structural studies of conotoxins'). The additional minor peaks could also be due to the formation of polymeric material.

Point 5. In figure 2C, co-elution of the selectively folded material with the random air oxidation. However, the background peaks do not look like the result of random air oxidation.

Response 5. It is possible that these peaks are due to minor contamination outside of the formation of additional isomers or the reasons stated in our response to 4. Nevertheless, the peaks of interest are the correct mass and co-elute with the selectively oxidized material. It is unclear to us as to how these additional peaks may impact the validity of the results.

Point 6. To convince me, authors should coelute three synthetic isomers with three native PnID from venom.

Response 6. As mentioned earlier, we agree that such an experiment would be ideal. However, it would be exceedingly difficult and costly to recollect the native venom for this experiment.

Point 7. In section 2.3, the authors identified the disulfide bond of synthetic PnID by reduction and alkylation. More importantly, authors should determine the disulfide bond linkage of native PnID purified from venom.

Response 7. As mentioned earlier, although this would be ideal, it would be exceedingly difficult and costly to recollect the native venom for this experiment

Point 8. The annotations of Figure 3 and Figure 4 mix up.

Response 8. Thank you for spotting this; this has now been fixed.

Point 9. Statistical analysis methods are required for pharmacological activity in Figure 5.

Response 9. The following has been added to the description in figure 5 “Statistical analyses were carried out using GraphPad Prism 6.0 (GraphPad Scientific, San Diego, CA, USA). IC50 values for uptake inhibition were calculated based on non-linear regression analysis.” This statement is also present in the methods section.

Point 10. Authors should modify and improve the quality of the Figures in Supplementary Materials.

Response 10. We agree that the quality of these figures is lower than those within the paper, the short deadline provided by the journal for submitting revisions did not allow us to redraw these figures to make them more aesthetically appealing. In addition, we believe that their current resolution is sufficient for their role as supplementary figures rather than key figures in the paper.

Point 11. In the disulfide bond determination by reduction and alkylation section (line134), the authors identified sequence using matrix-assisted laser desorption/ionization mass spectrometry (PSD-MALDI-MS), but I do not see any description of this experiment in Materials and Methods section. Moreover, the authors identified the secondary ion fragments using tandem mass spectrometry (Supplemental Fig. 3), the authors did not analyze the data clearly.

Response 11. Thank you for the comment. The header for this section of the methods was missing and is now included in line 327. We also agree that a synopsis of the data in this supplemental figure is not explained as clearly as the data included in the manuscript as it is supplemental and provided for reader interpretation.

Reviewer 2 Report

The work presented in the manuscript is of clear interest:  the authors had  information  about the presence  of three isomers of the conotoxin PnID, here  they prepared by regioselective synthesis  three respective forms differing in the disposition of the two  intramolecular disulfide bonds, determined their H-NMR structures  and demonstrated that   only the ribbon  form had the inhibitory activity against the norepinephrine transporter. It should be mentioned that the family of the known kappa-conotoxins is very limited and PnID  is the first kappa-conotoxin isolated from the  Conus penaceus snail. Among major  advantages of this work  should be mentioned  the synthesis of isomers and determination of their spatial structure. Speaking about possible application of  this conotoxin as a drug, the authors mentioned that the related conotoxin MrIA showed good activity against the rat norepinephrine  transporter, but its clinical studies were stopped because it did not show the expected activity against the human transporter – since the authors  also  tested the PnID activity  only against the rat protein, testing against the human transporter   would be desirable.

Minor critical remark: on p.8 lines 199-200 they  write  that “ earlier in our endeavor  we determined several native isomers of  the novel peptide PnID” – it is not clear to me whether this has been already published (even in a very short form) or in the present manuscript it is described  for the first time.

Author Response

Point 1. It should be mentioned that the family of the known kappa-conotoxins is very limited and PnID is the first kappa-conotoxin isolated from the Conus penaceus snail.

Response 1. Thank you for the comment and we completely agree. We believe the reviewer means to refer to chi conotoxins instead of kappa. The paper includes the following statement on lines 65-66 to address this concern: “PnID is also the first χ-conotoxin to be isolated from Conus pennaceus.”

Point 2. Among major advantages of this work should be mentioned the synthesis of isomers and determination of their spatial structure.

Response 2. Thank you again for the insightful comment. We have modified lines 58-59 to now state “Although PnID was previously identified by genetic analysis[4,8], this is the first report of the pharmacology, chemical synthesis and three dimensional structure of this peptide” to address this concern.

Point 3. Speaking about possible application of this conotoxin as a drug, the authors mentioned that the related conotoxin MrIA showed good activity against the rat norepinephrine transporter, but its clinical studies were stopped because it did not show the expected activity against the human transporter – since the authors also tested the PnID activity only against the rat protein, testing against the human transporter would be desirable

Response 3. We are also in agreement with this statement. Unfortunately, we do not currently have a cell line capable of expressing the human receptor. It is our hope that we could develop such a cell line in the future and further assess this conotoxin and analogs of this conotoxin against the human receptor in a separate publication.

Point 4. Minor critical remark: on p.8 lines 199-200 they write that “ earlier in our endeavor we determined several native isomers of the novel peptide PnID” – it is not clear to me whether this has been already published (even in a very short form) or in the present manuscript it is described for the first time.

Response 4. This work has not previously been published in a peer reviewed journal and is being described here for the first time. As per the reviewer comment we believe the words “earlier in our endeavor” are misleading. This wording was originally chosen because the authors began the project to discover a new conotoxin and did not expect to find the additional native isomers. We have now rephrased this to “At the beginning of this project” rather than “early in our endeavor”.

Reviewer 3 Report

Espiritu et. al observed that a peptide, PnID, from Conus pennaceus venom, homologous to χ-conotoxins, has three natural isoforms (A, B and C), which have a different retention time in LC-MS, and which become a single form upon treatment of the venom with a reducing agent. PnID is a peptide of 12 amino acids containing 4 cysteines, which theoretically can form 3 different combinations of 2 disulphide bonds, resulting in three isoforms called globular, ribbon and beaded.

To find out whether the three natural isoforms of PnID are due to different couplings of the cysteines in forming disulphides, the authors synthesised the peptide by forming the two bonds via a regioselective oxidation, using two different protective groups of the cysteine side chains, and randomly, in air (assuming that the native folding occurs this way). By comparing the retention times in LC-MS of the regioselectively folded peptides with the air-folded ones, they observed that the air-folded peptide has only two of the three possible isoforms, the globular, and the ribbon.

They then characterised the pharmacological activity of the three isoforms of PdID, obtained by regioselective oxidation, and observed that only one of the three isoforms, B (ribbon), has full rat norepinephrine transporter inhibition activity, while A (globular) has partial activity, and C (beaded) has no activity.  

This work is interesting both because it analyses a new conotoxin, produced by a different species of conus, and because it is the first case of a conotoxin that has three native isomers due, in at least two cases, to a different coupling of the cysteines in forming the disulphide bonds. The fact that only one of the isomers has the expected pharmacological activity suggests that the other two isoforms may have other pharmacological activities that are yet unknown.

Remarks:

-          Authors should explain where the terms globular, ribbon and beaded come from, in the introduction or when they introduce the terms for the first time (excluding the abstract)

-          The authors should point out that the beaded form includes a vicinal disulphide which, although it exists in some cases, is very rare in nature (PMID: 28336403).

-          In the discussion (lines 210-214), when the authors talk about the possible structure of the third isomer they should also consider that, with the same coupling of the cysteines in forming the disulfide bonds, different structural isomers can be formed (Daly and Craik, IUBMB Life 2009 'Structural studies of conotoxins'), i.e. for the conformation of the five dihedral angles of the cystine residue, or for the cis trans isomerization of proline, and that the native folding, in the presence of chaperons, could favour one shape over another.

Minor: the captions of Figures 3 and 4 have been switched

Author Response

Point 1. Authors should explain where the terms globular, ribbon and beaded come from, in the introduction or when they introduce the terms for the first time (excluding the abstract)

Response 1. Thank you for this suggestion. The text has been modified on lines 49- 54 to now state “…and the preferential formation of unique disulfide connectivity[6,7]. For conotoxins with four cysteines, three possible disulfide connectivity patterns are hypothetically achievable: the globular connectivity (pairings between C1-3 and C2-4), the ribbon connectivity (pairings between C1-4 and C2-3), and the beaded connectivity (pairings between C1-2 and C3-4; also see Fig 2). χ-Conotoxins are known for their preferential formation of a ribbon connectivity.” to address this concern.

Point 2. The authors should point out that the beaded form includes a vicinal disulphide which, although it exists in some cases, is very rare in nature (PMID: 28336403).

Response 2. The text has been modified at lines 222 to 225 to include the following sentence to address this concern: “The native presence of a beaded isomer would be unusual as this isomer contains energetically costly vicinal disulfide bonds. However, despite the higher energy cost of producing native vicinal disulfides, such bonds have been reported previously in nature albeit rarely [13].”

Point 3. In the discussion (lines 210-214), when the authors talk about the possible structure of the third isomer they should also consider that, with the same coupling of the cysteines in forming the disulfide bonds, different structural isomers can be formed (Daly and Craik, IUBMB Life 2009 'Structural studies of conotoxins'), i.e. for the conformation of the five dihedral angles of the cystine residue, or for the cis trans isomerization of proline, and that the native folding, in the presence of chaperons, could favour one shape over another.

 Response 3. Thank you for the insightful comment. To address this concern the text has been modified at lines 225-228 to include the following statement: “Other possibilities to the identity of this third isomer could be the inclusion of additional dihedral angles of the cysteine residues or the cis isomerization of proline, either of which may be favored in the native state due to the presence of chaperones.

Point 4. Minor: the captions of Figures 3 and 4 have been switched

Response 4. Thank you very much for catching this. The captions have now been corrected.

Round 2

Reviewer 1 Report

Revised by the authors,I agree that the article can be published in its current version.